# Patient and family engagement in infection prevention in the context of the COVID-19 pandemic: defining a consensus framework using the Q methodology – NOSO-COVID study protocol

Nathalie Camille Clavel ,[1] Mélanie Lavoie-Tremblay,[1] Alain Biron,[2] Anaick Briand,[2] Jesseca Paquette,[1] Laurence Bernard,[3] Carol Fancott,[4] Marie-Pascale Pomey,[5] Vincent Dumez[6]

For numbered affiliations see end of article.

**Correspondence to**
Dr Nathalie Camille Clavel; nathalie.clavel@mail.mcgill.ca

## ABSTRACT

**Introduction** Healthcare-associated infections are an important patient safety concern, especially in the context of the COVID-19 pandemic. Infection prevention and control implemented in healthcare settings are largely focused on the practices of healthcare professionals. Patient and family engagement is also recognised as an important patient safety strategy. The extent to which patients and families can be engaged, their specific roles and the strategies that support their engagement in infection prevention remain unclear. The overarching objective of the proposed study is to explore how patients and families can effectively be engaged in infection prevention by developing a consensus framework with key stakeholders.

**Design and methods** The proposed study is based on a cross-sectional exploratory study at one of the largest university hospitals in North America (Montreal, Canada). The targeted population is all healthcare professionals, managers and other non-clinical staff members who work on clinical units, and the in-patients and their families. The study is based on Q methodology that takes advantage of both quantitative and qualitative methods to identify the consensus among the various stakeholders. This exploratory Q research approach will provide a structured way to elicit the stakeholders' perspectives on patient and family engagement in infection prevention.

**Ethics and dissemination** The research ethics board approved this study. The research team plans to disseminate the findings through different channels of communication targeting healthcare professionals, managers in healthcare settings, and patients and family caregivers. The findings will also be disseminated through peer-reviewed journals in healthcare management and in quality and safety improvement.

## INTRODUCTION

Healthcare-associated infections (HAIs) are the most frequent complications for patients

### STRENGTHS AND LIMITATIONS OF THIS STUDY

⇒ The study is based on Q methodology, which is relevant for identifying the consensus among stakeholders for a set of strategies and actions to be implemented in healthcare settings.
⇒ The proposed Q study takes advantage of mixed research methods to understand the perspectives of different key stakeholders on patient and family engagement in infection prevention.
⇒ The study will be conducted with key stakeholders who play a role in infection prevention and control (patients, family caregivers, healthcare professionals, managers and other non-clinical staff members).
⇒ The study will unfold within a single healthcare organisation, which may limit the generalisability of the findings.
⇒ This study will be conducted in different clinical units to compare the different contexts of care.

receiving healthcare. The threat requires increased vigilance in healthcare settings. Internationally, 5%–15% of patients admitted to hospitals acquire an infection.[1] In Canada, HAIs remain particularly persistent with more than 200 000 Canadians being infected every year while receiving healthcare.[2] HAIs are infections that are acquired in patients in settings where care is delivered, including healthcare settings (eg, hospitals or long-term facilities).[3] Examples of bacterial HAIs that are commonly found in healthcare settings include methicillin-resistant *Staphylococcus aureus* (MRSA) infections, vancomycin-resistant enterococci (VRE) infections and *Clostridium difficile* infections.[1 4] These infections have major health-related and financial consequences for health systems, including

higher mortality and morbidity rates and increased lengths of stay at hospitals.[5–7]

In the current context of the spread of SARS-CoV-2 in the community, with asymptomatic people spreading the virus, and a proportion of infected people being hospitalised, the incidence of HAIs is expected to increase in the coming months and years. One rapid review and meta-analysis showed that 44% of the confirmed cases of COVID-19 were acquired within healthcare settings.[8] Another paper highlighted the risk for inpatients with COVID-19 to have HAIs as secondary infections. Moreover, in adult inpatients diagnosed with COVID-19 in Wuhan, China, half of the non-survivors experienced a secondary infection.[9] SARS-CoV-2 is highly contagious and can be rapidly spread in healthcare settings if no specific prevention and control measures are implemented and sustained over time. Thus, the involvement and engagement of patients and their families is needed to prevent spread.

However, to date, infection, prevention and control (IPC) strategies implemented in healthcare settings have largely focused on the practices of healthcare professionals (HCPs), (eg, nurses, physicians, allied health professionals) and other staff members (eg, housekeepers, transport attendants, unit managers), without defining how patients and their relatives could be involved to reduce the infections. Patient and family engagement is widely recognised as a promising strategy for improving the quality of healthcare and patient safety.[10–13] In recent years, several studies have examined patient engagement in IPC.[14–17] In the literature, a consensus has been expressed that patients, family caregivers and health professionals should play joint roles in preventing HAIs. Nevertheless, the extent to which patients and families can effectively be engaged and their specific roles and responsibilities in IPC remain unclear.[18 19] The evidence suggests that patients may feel anxious or uncomfortable about asking about or getting involved in their safety.[20] Studies have also reported that health professionals often acknowledge patients as being vulnerable though not necessarily co-responsible for preventing infection transmission.[18] Finally, a few studies have identified existing strategies to engage patients and family caregivers in IPC,[18] though developing targeted strategies to enable patient and family engagement might contribute to enhance IPC activities.

In line with the recommendations of the World Health Organisation[21] and the action plan of the Quebec Ministry of Health and Social Services,[22] a Montreal university hospital implemented an interdisciplinary organisational auditing programme that aims to reduce HAIs. This nurse-led programme, implemented in 2016 in several clinical units focuses on increasing the compliance of: (1) hand hygiene, (2) appropriate glove use, (3) appropriate personal protective equipment use and (4) proper cleaning and disinfection of environment and equipment. To achieve these objectives, four main strategies were implemented: (1) staff training on IPC,

IPC tools and process improvements, (2) audits of hand hygiene, (3) glove use and disinfection of equipment and (4) communication reinforcement on IPC among staff using huddles.[23] Despite improvements in nosocomial infection rates (*C. difficile* decreased from 13 to 6.8 per 10.000 patient days, MRSA rate decreased from 26 to 12 per 10.000 patient days, and VRE rate decreased from 8 to 4.5 per 10.0000 patient days) and hand hygiene compliance among staff from 37% to 67%,[23] audits on clinical units showed that hand hygiene rates among patients and visitors remained low (under 50%). The strategies that were implemented specifically targeted the practices of HCPs,[23] without defining how patients and family caregivers could be involved to reduce the infections.

During the pandemic, healthcare organisations have implemented new measures that target patients and family caregivers visiting patients. Some of these measures have increased patient and family responsibilities in infection prevention (eg, routine hand hygiene, systematic wearing of masks), while other measures have limited the family caregivers' essential roles as partners (eg, prohibition or limitation of non-essential visits).[24 25] These restrictive policies were enacted quickly as the pandemic emerged, without input from all stakeholders (including patients, family caregivers, clinical and non-clinical staff members) who play a major role in IPC. Thus, creating a consensus framework will help to define the patients' and family caregivers' roles and responsibilities in IPC during a pandemic and identify the strategies for promoting and facilitating their engagement in this important area of patient safety.

### Aims and objectives

To address the gap, the overarching objective of the study is to explore how patients and families can effectively be engaged in IPC in the context of the COVID-19 pandemic. Specifically, the study is based on the following objectives:

Objective 1: to measure and understand patients' and families' preferences regarding their engagement in IPC.

Objective 2: to measure and understand the preferences of clinical and non-clinical staff on patient and family engagement in IPC.

Objective 3: to identify a consensus framework for the roles and responsibilities of patients and families in IPC and strategies to facilitate their engagement.

## METHODS AND ANALYSIS
### Project design and research approach

The proposed study uses a cross-sectional exploratory approach. The targeted population is comprised of all clinical and non-clinical staff members who work in clinical units of the participating healthcare organisation that have implemented the interdisciplinary organisational IPC programme, and patients hospitalised in these units and their families or other caregivers. The

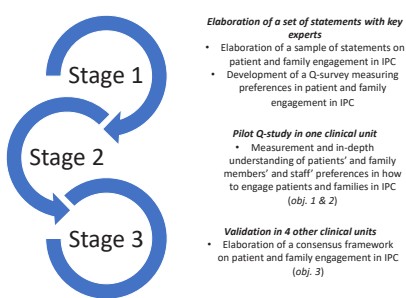

**Figure 1** Study stages.

study is based on the Q methodology,[26] an exploratory research approach that provides a structured way to elicit the stakeholders' perspectives on the issue by using both qualitative and quantitative methods. It offers an innovative alternative to traditional methods for statistically analysing a variety of perspectives that can lead to the development of a consensus on a set of actions, interventions or policies for implementation.[26 27]

## Project setting and stages

The research project will take place at an academic health network that is one of the most modern and largest bilingual university networks in North America (2019–2020: more than 36 000 admissions, around 30 000 surgeries, more than 5000 nurses and patient attendants, 1300 physicians and around 4000 other professionals). The healthcare organisation provides tertiary and quaternary care to the population of Montreal and Quebec. The project will be divided into three stages that have been identified as feasible by key stakeholders at the participating healthcare organisation.

The Q study is divided into three successive stages (figure 1). The different subsections in the methods section will be presented for each stage successively.

## Conceptual framework

The study relies on a theoretical framework adapted from Pomey *et al*[28] on patient and citizen engagement in healthcare systems and on the work of Arnstein[29] on citizen participation in public policy (figure 2). The framework describes the spectrum of patient and family engagement in infection prevention from paternalism to partnership (coresponsibility). The higher level of engagement involves supporting a shared responsibility among patients, families, and clinical and non-clinical staff members towards infection prevention. At the extreme opposite of the spectrum, mechanisms can be implemented to ensure that patients and families comply with infection prevention measures or are informed about infection prevention practices in hospital settings.

## Stage 1: elaboration of a set of statements with key experts (fall 2020–winter 2021)
### Objective

The preparatory phase of the study will consist of elaborating on a sample of statements (Q set) on patient and family engagement in IPC that will be used to develop the Q-survey comprising around 40 items that will be administered to participants in the second stage (pilot Q study).

### Setting and participants

The participants will be comprised of key stakeholders involved in the interdisciplinary organisational programme focusing on IPC and managers and coordinators responsible for patient and family engagement at the participating healthcare organisation. Stakeholders involved in IPC will be recruited through the steering committee of an IPC interdisciplinary organisational programme responsible for reflecting, assessing and orientating appropriate practices in IPC. The IPC programme steering committee brings together HCPs (including clinical managers), patients as partners and members from different departments who have a key role regarding IPC (ie, nursing, medical affairs, infection

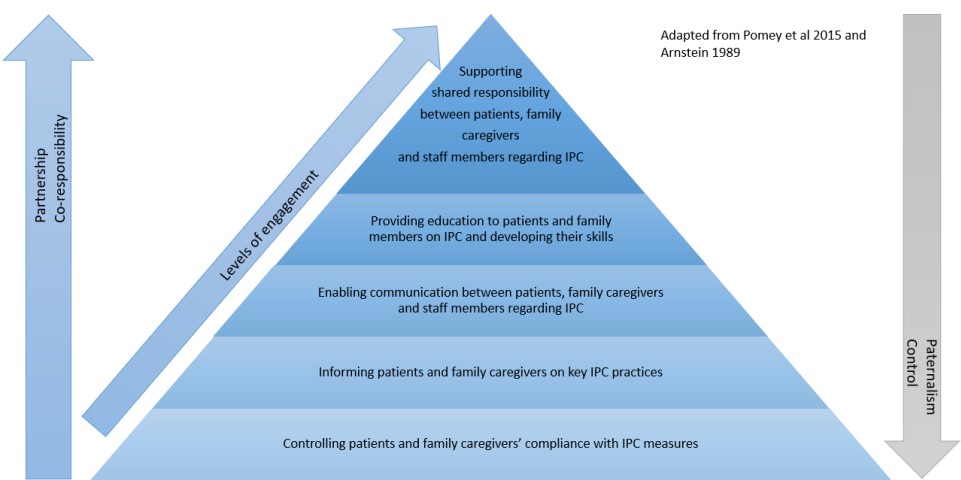

**Figure 2** Conceptual framework. IPC, infection, prevention and control.

control, housekeeping and quality). The IPC programme has been ongoing since 2016 and more than 30 units have implemented the strategies targeting improvement in IPC practices within the organisation. The principal investigator will present the study during 1 monthly meeting of the steering committee and members will be invited to participate in the focus group.

In addition, managers and coordinators of patient and family engagement will be recruited through the quality department that is responsible for implementing patient engagement practices within the organisation. The participants will be identified with the help of the quality department of the healthcare facility and invited to participate in a focus group by email.

We decided to conduct two separate focus groups, as potential participants represent two groups who might have different viewpoints; the members of the IPC steering committee are interested in IPC measures and strategies and the managers and coordinators of patient and family engagement are interested in strategies that foster patient participation in the healthcare facility. We seek to recruit a total of 10–12 participants and plan to organise a focus group with each group of stakeholders (5 experts per group: members of the steering committee on IPC and managers/coordinators of patient engagement). We adopted a strategic approach to participant recruitment as it is recognised in Q methodology[30]; we seek to recruit participants who may have various viewpoints to express that matters in relation to IPC and patient and family engagement (HCPs, other staff members, managers/coordinators, patients as partners). We think that recruiting 10–12 participants will ensure this diversity in viewpoints at this stage.

### Data collection and analysis

Statements on patient and family engagement in IPC will be drawn from two sources. First, a literature review of scientific papers and grey literature will be conducted to highlight key findings in the area of patient engagement in IPC, including the specific context of the COVID-19 pandemic. Second, the qualitative data derived from two focus groups conducted with key experts in IPC and patient and family engagement will be used. Data from these two sources will be integrated to draw a list of approximately 40 statements that represent various perspectives on patient and family engagement in IPC.

### Stage 2: pilot Q study in one clinical unit (spring and summer 2021)

#### Objective

This stage will help to measure and understand the preferences of patients, family caregivers and staff members on patient and family engagement in IPC (objectives 1 and 2).

This stage will help to identify different groups of perspectives regarding the roles of patients and their families in IPC and the strategies/actions that can be implemented to enhance the preferred roles.

### Setting and participants

This stage will be conducted on one of the clinical units (adult surgery) that has implemented the interdisciplinary programme focusing on IPC. This unit has been chosen for two reasons: (1) the unit has shown low hand hygiene compliance rates from patients and visitors before the pandemic (37% in 2019) and (2) patients hospitalised in this unit are immunosuppressed and they need greater protection from infections. The unit has been identified and contacted with the help of the nursing department of the participating healthcare facility. The targeted total sample will be 30 participants: 15 members of the clinical unit (health professionals, managers and other non-clinical staff members) and 15 patients or family caregivers. The recruitment of patients, family caregivers, and staff members will be carried out by the nurse manager and the assistant nurse manager of the unit.

#### Inclusion criteria

All staff currently working in the clinical unit (HCPs, managers and other staff members) and who are willing to participate will be included in the study. Among HCPs, we seek to recruit surgeons, nurses, nurse/assistant nurse managers, physiotherapists and beneficiary attendants. Among other staff members, we seek to recruit housekeepers, clinical managers, unit receptionists. Patients who have been hospitalised or relatives whose loved ones have been hospitalised on these units will be considered for inclusion if they meet the following conditions: (1) 18 years or older; (2) have been hospitalised within the 2 months before data collection to limit memory biases; (3) speak and read French or English; (4) have no serious cognitive or speech impairments, or any other conditions that would limit their ability to provide informed consent, and (5) have access to the internet.

### Data collection

In the pilot Q study, data collection will be based on two methods used in Q methodology (online supplemental Q survey and interviews), which will be done remotely because of the COVID-19 pandemic.

All participants will be invited to complete the Q online survey developed in stage 1. The Q survey will consist of socio-demographic information questions and around 40 statements that participants will have to sort and rank according to a two-step sorting and ranking process specific to the Q methodology. This first step will consist of sorting each statement into three categories (agree, neutral, or disagree), and then a second step will ask participants to prioritise statements by ranking them in five categories (−2 completely disagree, −1 disagree, 0 neutral, +1 agree, +2 completely agree). A forced-choice distribution of statements will allow the participants to add a limited and fixed number of statements for each category (4 items in +2/−2, 8 items in +1/−1, 16 items in 0). The Q survey will be hosted on a specific platform (QsorTouch) that has been developed for Q methodologist

researchers to easily and securely administrate Q online surveys.

Postsorting/ranking interviews will be conducted with participants after they have completed the survey and after the researcher has reviewed each participant's Q sort (ranking of statements). As part of the interviews, participants will be asked to explain the rationale of ranking statements in the most extreme scales (ie, −2 or +2) and some of the statements ranked in the other scales (0 and −1/+1). Phone interviews will last approximately 30 min and will be audiorecorded.

## Data analysis
### Quantitative data from Q surveys
At stage 2, data analysis will consist of comparing and grouping by similarity all of the Q sorts that are collected. The number of Q sorts will be equal to the number of participants (n=30). A by-person factor analysis will be undertaken to measure the intercorrelation of all of the gathered Q sorts to identify a group of persons who ranked the order of statements in a very similar fashion (Q factors).[30] A factor in Q methodology represents a group of persons who share a similar perspective or viewpoint about patient and family engagement in IPC. Grouping, comparing and summarising the Q sorts will be performed using data-reduction techniques, including principal component analysis (PCA). The analysis will be done using PQMethods software, which is dedicated to analysing Q data. PCA consists of reducing the data to a few factors. The reduction in PCA will be done according to two steps: extraction and rotation. Extraction consists of summarising all of the participants' responses into a few representative responses—the Q perspectives. Rotation is used to have a clearer structure of the results, including an understanding of the percentage of variability explained by the Q perspective.[30] In this study, an example of a Q perspective could be participants in favour of controlling patients and family members' compliance with IPC practices. Factor loadings will be calculated to explain the relationship between each participant and factor (Q perspective). Z scores will also be examined to 'give more precision about how strongly engaged each perspective is with each item'.[26]

### Qualitative data from interviews
Transcriptions from the interviews will be analysed using QDA Miner. A thematic analysis will be performed based on our conceptual framework (figure 2) and following Miles and Huberman methods for qualitative data analysis (Miles & Huberman, 2014).

### Integration of quantitative and qualitative data
The interpretation of the results will be based on both quantitative data and qualitative data. Each factor will be described in detail and given a meaningful label to indicate what similar perspectives are about; for example: 'controlling patients and family members' IPC practices' or 'supporting shared responsibility among patients,

family members, and staff members for IPC.' Each perspective will reveal and describe the preferences of participants regarding patient and family engagement in IPC. The in-depth qualitative data will allow for an understanding of the rationale of each perspective.

## Stage 3: validation in four other clinical units (fall 2021–winter 2022)
### Objective
This stage of the study is designed to validate the pilot Q findings from a larger sample of clinical units. Since the pilot Q study is based on a small sample, it will not be possible to generalise the findings to a larger population (eg, other clinical units). This can be done by combining a Q study methodology with an R methodology (traditional survey method).[31]

### Setting and participants
The four participating units at the MUHC (McGill University Health Center) will be randomly selected. All staff members (clinical and non-clinical) and patients and family caregivers will be invited to participate in the study. All potential participants will be given a package containing a letter of introduction describing the study, a demographic survey and a link to the survey that will be administrated online.

In stage 3, the targeted survey sample of participants is 200 participants, equally distributed among the four clinical units that will be randomly selected: 70 patients, 30 family caregivers, 70 HCPs (including managers), 30 non-clinical staff members (housekeepers, transport attendants, etc). The inclusion criteria for the patients and family caregivers will be the same as for stage 2.

### Data collection
#### Creation and administration of a traditional survey
To create a traditional survey, several highly and weakly ranked distinctive statements from each Q perspective measured at stage 2 will be included and presented as Likert items (a 5-point Likert scale). This technique is the most commonly used for combining Q methodology and survey methods.[31 32] The participants in each clinical unit will be invited to complete the online survey.

### Data analysis
The survey data will be entered in MS-Excel and then exported and analysed using SPSS software (V.28.0.1).

Any returned surveys that are missing one or more responses will be excluded from the analysis. For each sample (each clinical unit), we will perform descriptive statistics, calculating mean scores, SDs and the percentage of agreement for each item and the overall scale. High score items shared in the four samples will be identified to develop a consensus framework on the roles of patients and families and strategies that enhance their engagement in IPC. In addition, item scores will be compared across the samples (clinical units) for type of respondents (patients, relatives, HCPs, managers and other non-clinical staff members), and across different

demographic characteristics (age, gender and level of education). The comparisons will help to identify shared preferences across units, in terms of patient and family roles and strategies to enhance their engagement in IPC. The comparisons will also help to show differences between the participants and clinical units in terms of preferences for engaging patients and families in IPC.

### Patient and public involvement

Patient advisors will be involved in the pilot testing of the online surveys and in the different stages of the study to have their inputs and feedback on the different tools used for data collection and the preliminary results (stages 1, 2 and 3). One of the collaborators for this study is a patient expert who is a leading member of a reference Canadian centre in patient and family engagement.

### DISCUSSION

To date, little is known about how patients and families could be effectively engaged in IPC, an important area of patient safety.[18 33 34] Patient and family engagement in infection prevention remains unclear and is subject to barriers that have been studied previously.[18 35 36] Moreover, the COVID-19 pandemic has raised important trade-offs between patient safety/IPC practices on one hand and patient and family engagement as essential partners in the other hand.[37–39] Developing targeted strategies to overcome the barriers associated with patient engagement is important for patient engagement to become a core component of IPC programmes. Engaging patients in IPC can help reduce the burden of HAIs in clinical units and healthcare settings. Because of the complex reactions and barriers regarding patient engagement in patient safety, this study will help define the 'optimal' roles of patients in IPC and targeted strategies that are acceptable for stakeholders, including patients, family caregivers, HCPs, managers and other non-clinical staff members.

The proposed Q study is based on an innovative methodological approach (the Q methodology) that takes advantage of qualitative and quantitative methods to explore the subjectivity of perceptions in a statistical way. Using this methodology is particularly relevant to understand in a structured and clear way the various perspectives on patient and family engagement in IPC. To date, current studies on patient engagement in IPC have focused on qualitative methods that provide few insights on consensus strategies among the many actors in the area of patient engagement in patient safety and IPC.

### Potential implications for healthcare settings, patients and families

This study can contribute to improving IPC programmes that are currently focused on HCP best practices in IPC, including hand hygiene. During this time of pandemic, the importance of engaging all stakeholders in IPC processes is paramount. This research will help develop additional and targeted strategies to better engage patients and their families in patient safety, in the important area of infection prevention. Based on several clinical units, the Q study will also help to elaborate tailor-made strategies of patient engagement in IPC in different clinical contexts.

### ETHICS AND DISSEMINATION
### Ethical and data protection considerations

Written consent will be obtained from all participants before each interview. Ethical approval for the current study was obtained from the McGill University Health Centre Research Ethics Board approved this study (Patient and Family Engagement/2020-6404). Electronic data, that is, information and consent forms, quantitative data from surveys, digital recordings and interview transcriptions will be password protected and only accessible by members of the research team. Transcription data will be anonymised and audiorecorded interviews destroyed after they are transcribed.

### Outputs and dissemination

This study has been awarded funding by the Social Sciences and Humanities Research Council as part of the Partnership Engage Grants for which we developed a partnership with Healthcare Excellence Canada and a collaboration with the Centre of Excellence on Partnership with Patients and the Public (CEPPP). The pandemic has challenged many of the principles of patient partnership in care and the inclusion of family caregivers as essential partners in care, especially with the restrictive visiting policies. In Canada and abroad, the pandemic reminds us of the essential presence of family caregivers to continue to provide moral and physical support particularly to vulnerable patients such as older adults and children. In this context, Healthcare Excellence Canada and CEPPP are particularly committed to finding effective ways to better involve patients in IPC and to reintegrate the family presence in healthcare settings. Such a framework will be especially useful for Healthcare Excellence Canada in their 'Better Together' programme that focuses on reintegrating family caregivers as essential partners in care. The framework could also be used as a tool designed for health professionals and managers, and patients and family caregivers to help guide and support inclusive engagement initiatives in IPC in the context of a pandemic and beyond.

As part of our partnership and collaboration, various knowledge mobilisation activities, including knowledge synthesis, dissemination and exchange and cocreation with knowledge users and partners will be codeveloped. Dissemination activities and approaches will leverage the networks and existing structures of our partners, including Healthcare Excellence Canada and CEPPP, to broadly share the cocreated framework to health decision-makers, HCPs, patients and family caregivers. The research findings will be presented and submitted

to international quality and safety management and improvement conferences and journals.

**Author affiliations**
[1]Ingram School of Nursing, McGill University Faculty of Medicine, Montreal, Québec, Canada
[2]McGill University Health Centre, Montreal, Québec, Canada
[3]Faculty of Nursing, University of Montreal, Montreal, Québec, Canada
[4]Patient Engagement & Partnerships, Healthcare Excellence Canada, Ottawa, Ontario, Canada
[5]Universite de Montreal, Montreal, Québec, Canada
[6]Centre of Excellence on Partnership with Patients and the Public, Montreal, Québec, Canada

**Acknowledgements** We thank the Social Sciences and Humanities Research Council for providing research funding for this study. The authors also thank Glen Wheeler for his contribution to the editing of this manuscript.

**Contributors** The study concept and design were conceived by NCC with substantial feedback from ML-T, ABi, and ABr. NCC and JP will conduct the data collection and analysis. NCC prepared the first draft of the manuscript, and all authors (NCC, ML-T, ABi, ABr, JP, LB, CF, M-PP and VD) provided edits and critiqued the manuscript for intellectual content. All authors read and approved the final manuscript.

**Funding** This study has received funding from the Social and Sciences Humanities Research Council (Government of Canada) through the Partnership Engage Grant-a special initiative for COVID-19 (SSHRC 1008-2020-1062). This study is also supported by a postdoctoral scholarship to Nathalie Clavel from the Fonds de Recherche du Québec- Santé. Finally, research funds were provided by the McGill University Health Center and the McGill Nursing Collaborative for Education and Innovation in Patient-Centred and Family-Centred Care (Newton Foundation/McGill Faculty of Medicine) (N/A).

**Competing interests** None declared.

**Patient and public involvement** Patients and/or the public were involved in the design, or conduct, or reporting, or dissemination plans of this research. Refer to the Methods section for further details.

**Patient consent for publication** Not applicable.

**Provenance and peer review** Not commissioned; externally peer reviewed.

**ORCID iD**
Nathalie Camille Clavel http://orcid.org/0000-0002-0438-6655

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
