## [Reviewer comments · BMJ Open]

ARTICLE DETAILS

TITLE (PROVISIONAL)	Patient and family engagement in infection prevention in the context of the COVID-19 pandemic: Defining a consensus framework using the Q methodology - NOSO-COVID study protocol
AUTHORS	Clavel, Nathalie; Lavoie-Tremblay, Mélanie; Biron, Alain; Briand, Anaick; Paquette, Jesseca; Bernard, Laurence; Fancott, Carol; Pomey, Marie-Pascale; Dumez, Vincent

VERSION 1 – REVIEW

REVIEWER	Peter Wilson University College London Hospitals NHS Foundation Trust, Microbiology
REVIEW RETURNED	16-Dec-2021

GENERAL COMMENTS	Infection prevention in the household is important in the control of covid-19 and a topic worthy of further investigation. Patient behaviour is a major factor in acquisition of infections including Covid-19. Public engagement in infection prevention has affected the severity of pandemic waves. Initial statements are obtained from a panel of stakeholders but no patient representatives. Inclusion criteria for the second stage that involves patients excludes those with no internet access. Hence the most vulnerable may be excluded. Methodology looks to be appropriate, focussing on stakeholder views rather than actions. Stages 1&2 are scheduled to have happened already. The initial panel involves patient coordinators, but it is not clear if these are themselves patient representatives. The larger Stage 2 panel has equal membership of staff and patients. The sorting of statements according to level of agreement is used widely. Patient advisors are stated to be involved in each stage, presumably with potential ability to alter design. Covid-19 provides specific challenges for managing infection control in the household. The public have received high levels of infection prevention advice over the last two years so it will be compliance rather than knowledge that would be expected to be the determinant for Covid-19. For other pathogens knowledge is likely to be more unreliable. There are clear plans for dissemination of results.
---

REVIEWER	Ermira Tartari University of Malta, Faculty of Health Sciences
REVIEW RETURNED	28-Jan-2022

GENERAL COMMENTS	Thank you for submitting this study protocol. Patient engagement and involvement in their care in terms of infection prevention is a topic that deserves further research. Kindly see some points below that would require some further clarification
--

	Suggest the use of IPC acronym for infection prevention and control instead of IPAC since this is used broadly Introduction: page 4 line 41-45 you provide rates for improved hand hygiene but you do not do the same for stating improved infection rates, provide infection rates also Project Setting and Stages 'The research project will take place at an academic health network that is one of the largest in North America'. – you state this is the largest centre in North America, please identify how large the centre is? Can you elaborate on how the following objective will be conducted? 'Objective. The preparatory phase of the study will consist of elaborating on a sample of statements (Q set) on patient and family engagement in IPAC that will be used in the pilot Q study.' Please identify precisely who the key stakeholders are and how will they be identified 'The participants will be comprised of key stakeholders involved' Explain in detail the phases of recruitment and how this will be performed "...managers and coordinators of patient and family engagement will be recruited through the quality department that is responsible for implementing patient engagement practices within the organization" "We seek to recruit a total of 10 participants and plan to organize a focus group with each group of stakeholders (five experts per group: members of the steering committee on IPAC and managers/coordinators of patient engagement)." Explain the reason for aiming to recruit 10 participants, on what is the decision based? Identify the process for the focus group discussion, how did you decide that 2 focus groups will be conducted? "Second, the qualitative data derived from two focus Groups" Setting and participant "...are immunosuppressed and then need greater protection from infections" they instead of then Provide details of the study sample population – for example who is included in the group of health professionals, or in other staff members? Provide further clarification It is not clear how the wards/department are recruited/ selected for inclusion in the study Explain why such a long hospitalisation is needed? This might limit inclusion of patients, since few patients are hospitalised for so long, what was the rationale for doing this? "(2) have been hospitalized on the unit for two months" Is the interview/survey guide available? Will this be in French and English? "The Q survey will consist of socio-demographic information questions and 40 to 60 statements"
--	---

VERSION 1 – AUTHOR RESPONSE

Answer to Reviewer 1:

Comment #1: Initial statements are obtained from a panel of stakeholders but no patient representatives.

Initial statements were identified from the panel of key stakeholders that also include patients as partners. See page 7 for more details on key stakeholders included.

Comment #2: Inclusion criteria for the second stage that involves patients exclude those with no internet access. Hence the most vulnerable may be excluded.

We are conscious that the criteria might exclude the most vulnerable patients (elderly patients for example), but during the pandemic, the data collection had to be done remotely and by respecting the social distancing measures, and the only way to administrate the Q-survey (second stage) was online following a two steps process of completion.

Comment #3: The initial panel involves patient coordinators, but it is not clear if these are themselves, patient representatives.

During stage 1, patients as partners will be included, patients as partners are members of the IPC steering committee through which participants will be identified. We added more details on how key participants at this stage will be chosen on pages 7 and 8.

Answer to Reviewer 2:

Comment #1: Suggest the use of IPC acronym for infection prevention and control instead of IPAC since this is used broadly

We used IPC acronym throughout the manuscript instead of IPAC, as suggested.

Comment #2: Introduction, page 4 lines 41-45 you provide rates for improved hand hygiene but you do not do the same for stating improved infection rates, provide infection rates also
As suggested, we added details on the decreasing rates of three major HAIs.

Comment #3: Project Setting and Stages: 'The research project will take place at an academic health network that is one of the largest in North America'. – you state this is the largest centre in North America, please identify how large the centre is?

We have added more details on how large the center is, see page 6.

Comment #4: Can you elaborate on how the following objective will be conducted?

'Objective. The preparatory phase of the study will consist of elaborating on a sample of statements (Q set) on patient and family engagement in IPAC that will be used in the pilotQ study.'

We elaborated on the objective, see page 7.

Comment #5: Please identify precisely who the key stakeholders are and how will they be identified
'The participants will be comprised of key stakeholders involved.

We added a sentence that add details on how participants were identified and recruited.

We added some details on how the key stakeholders are described see page 7. Stakeholders involved in IPC will be recruited through the steering committee of the interdisciplinary organizational program responsible for reflecting, assessing, and orientating appropriate practices in IPC. The steering committee brings together healthcare professionals (including clinical managers), patients as

partners, and members from different departments who have a key role regarding IPC (i.e., nursing, medical affairs, infection control, housekeeping, and quality).

Comment #6: Explain in detail the phases of recruitment and how this will be performed
“...managers and coordinators of patient and family engagement will be recruited through the quality department that is responsible for implementing patient engagement practices within the organization”

We added a sentence that provides details on how participants were identified and recruited.

Comment #7: “We seek to recruit a total of 10 participants and plan to organize a focus group with each group of stakeholders (five experts per group: members of the steering committee on IPAC and managers/coordinators of patient engagement).” Explain the reason for aiming to recruit 10 participants, on what is the decision-based?

We provided more details on our strategic approach to recruiting a total of 10-12 participants for stage 1 (see page 8).

Comment #8: Identify the process for the focus group discussion, how did you decide that 2 focus groups will be conducted? “Second, the qualitative data derived from two Focus Groups”

We also provided more details on the process and reasons for conducting two separate focus groups at stage 1 (see page 8).

Comment #9: Setting and participant
“...are immunosuppressed and then need greater protection from infections” they instead of then
As suggested, we replaced then with they

Comment #10: Provide details of the study sample population – for example, who is included in the group of health professionals, or in other staff members? Provide further clarification
We provided details of the study sample population, types of health professionals included or other staff members (see page 9).

Comment #11: It is not clear how the wards/department are recruited/ selected for inclusion in the study
The unit has been identified and contacted with the help of the nursing department of the participating healthcare facility.

Comment #12: Explain why such a long hospitalization is needed? This might limit the inclusion of patients since few patients are hospitalized for so long, what was the rationale for doing this? “(2) have been hospitalized on the unit for two months”
We seek patients that were hospitalized within the two months before the time of data collection to avoid a memory bias so that they can recall their hospitalization, we seek patients that have been hospitalized but with no specific criteria concerning the length of stay. Initial sentence.” have been hospitalized on the unit for two months prior to the time of data collection” replaced by “have been hospitalized with the two months before data collection to limit memory biases.

Comment #13: Is the interview/survey guide available?
We attached the Q survey (40 items) in an additional file. We also included one example of the personalized interview guide used for interviews following Q survey completion at stage 2.

Comment #14: Will this be in French and English?
It will be administrated in both languages. Participants will have the choice to complete it in French or English. Interviews as well will be conducted in English or French according to participants’ preferences.